# Finite-Time Synchronization Control for Quaternion-Valued Memristive Neural Networks by Halanay Inequality

1st Jing Ping
*School of Mathematics*
*China University of Mining and Technology*
Xuzhou, Jiangsu, China
pingjing@cumt.edu.cn

2nd Song Zhu
*School of Mathematics, JCAM*
*China University of Mining and Technology*
Xuzhou, Jiangsu, China
songzhu@cumt.edu.cn

*Abstract*—This paper investigates the finite-time synchronization (FTS) of quaternion-valued memristive neural networks (QVMNNs) with mixed time delays. Utilizing the Halanay inequality, the analysis of the FTS issue involves two crucial steps: the convergence of the error system from the initial state to the unit ball, and subsequently from the unit ball to the origin. This approach not only obviates the necessity for constructing intricate Lyapunov functionals but also simplifies the treatment of delay terms. Furthermore, to streamline the theoretical derivation process, the considered QVMNNs are addressed without any decomposition by introducing the 1-norm, 2-norm, and ∞-norm of quaternions. Based on these considerations, a series of algebraic criteria for FTS is formulated, alongside estimates of synchronization time under different initial conditions. Finally, numerical simulations are presented to validate the effectiveness of the theoretical findings and demonstrate the application of QVMNNs in secure communication.

*Index Terms*—Finite-time synchronization (FTS), Halanay inequality, quaternion-valued memristive neural networks (QVMNNs), non-decomposition

## I. INTRODUCTION

Memristive neural networks (MNNs) are a special type of neural networks (NNs) based on memristors. Since the resistance of the memristors varies depending on the current and voltage in the circuit [1]–[3], the memristor is known as an electronic component with memory. This unique characteristic provides memristors with distinct advantages for information storage and computation [4], [5]. Due to the high efficiency in pattern recognition, cognitive computing, and simulating human neural systems [6]–[8], there have been a plethora of impressive accomplishments in the realm of MNNs research [9]–[11].

Through the extension of imaginary numbers, quaternions were initially introduced by Hamilton in the 19th century as a means of providing more precise representation of rotational changes in 3-D space. Generally, a quaternion $x$ can be represented as $x = x^R + x^I i + x^J j + x^K \hbar$, where $i, j$ and $\hbar$ are imaginary units that adhere to Hamilton rules: $i^2 = j^2 = \hbar^2 = -1$, $ij = -ji = \hbar$, $j\hbar = -\hbar j = i$, $\hbar i = -i\hbar = j$. Until now, the quaternions have been proven to be invaluable in various domains, including computer graphics and robotic control. The introduction of quaternions into NNs has led to the emergence of a new type of NNs, known as quaternion-valued NNs (QVNNs). QVNNs boast unparalleled potential in processing intricate data sets and excelling in multi-classification tasks compared to real-valued and complex-valued counterparts [12]–[15]. Despite these advantages, the research pertaining to QVNNs has not been comprehensive owing to the intricate and non-commutative nature of quaternion algebra. Several scholars have employed decomposition method to study QVNNs: dividing the considered QVNNs into four real-valued or two complex-valued NNs in accordance with Hamilton rules. For example, by splitting QVNNs into four real-valued NNs, [15] delves into the issues of quasi-synchronization and Hopf bifurcation of fractional-order QVNNs, [16] addresses the issue of global exponential stability. Whilst this methodology facilitates the transmutation of quaternions into real numbers, it is concomitantly accompanied by incurrence of augmented computational intricacy. Therefore, non-decomposition method has gained favor among scholars in recent years [17]–[19]. For instance, by treating the quaternion-valued system as undivided entity without decomposition, [18] investigates the global Mittag-Leffler stability of fractional-order QVMNNs, and [19] studies the dissipativity of QVNNs.

The dynamic behaviors, exemplified by stability and synchronization, are critically important for practical applications [20]–[23]. Specifically, the synchronization of networks has found extensive employment not only in communication and power systems, but also in numerous other systems [24], [25]. Over the past few years, research in the area of synchronization issues related to MNNs has experienced rapid growth, leading to a noteworthy amount of scholarly attention [26]–[29]. Nonetheless, quasi-synchronization, asymptotic synchronization and exponential synchronization can only guarantee that the states of error system can be reduced to a very small range or achieved synchronization within an infinite period of time. Furthermore, the strive to advance precision, efficiency and practical applicability has brought the subject of finite-time synchronization (FTS) control to the forefront of academic investigation [30]–[33]. By applying an appropriate controller,

[31] proposes some sufficient criteria for FTS of coupled MNNs. In [32], a novel analysis method is introduced to investigate FTS of inertial MNNs with mixed time delays. [33] studies the FTS issues of impulsive MNNs under stabilizing, inactive, and destabilizing impulses.

Time delay is a prevalent occurrence, which may have a negative impact during the process of achieving the desired system states. Therefore, it is imperative to take into account the effects of mixed time delays when analyzing the dynamic behaviors of NNs. For the purpose of addressing these issues, the Lyapunov functional method is an important tool, which exhibits superiority for treating delayed terms, such as [34]–[36]. It is noteworthy that the differentiability of time delays is typically assumed for the Lyapunov functional method. Furthermore, some scholars choose to design controllers that contain time delays [37], [38], and some make the additional hypothesis that the activation function is bounded [39]. Notably, Halanay inequalities are employed as novel and effective tools for studying asymptotic stability or synchronization issues of delayed systems [40], [41], owing to their ability to provide more precise analysis outcomes in certain cases. Nevertheless, their potential value in addressing FTS issue has been overlooked.

Driven by the aforementioned discussions, this paper is dedicated to investigating the FTS control of QVMNNs with mixed time delays via Halanay inequality. The principal contributions can be summarized as follows: 1) Distinct differential inequalities are established for regions outside and inside the unit ball, respectively, by applying the Halanay inequality and finite-time stability theory. This approach facilitates the handling of delay terms. Moreover, in comparison to prior studies [35], [36] and [42], this approach removes the requirements for differentiability and $\dot{\tau}(t) < 1$, as well as the need for constructing complex Lyapunov functionals. 2) The controller designed in this paper solely encompasses linear and sign function expressions, devoid of time delay terms. Contrasting with the controller in [42], it demonstrates simplicity, ease of manipulation, and obviates switching. 3) Based on the improved 1-norm, 2-norm, and $\infty$-norm of quaternion, a series of algebraic criteria for achieving FTS of QVMNNs is proposed without any decomposition. Compared with some previous results [15] and [16], this method can avoid the complex operation of the decomposition method, enrich the theories of applying non-decomposition method to study QVNNs, and provide more choices of control gains for achieving FTS of QVMNNs.

The paper is structured as follows. In Section II, the considered QVMNNs models and the relevant preliminary information are introduced. The FTS criteria for QVMNNs under 1-norm, 2-norm, and $\infty$-norm are shown in Section III. To show the effectiveness of these criteria, numerical simulations are presented in Section IV. Finally, the main work is reviewed and summarized in Section V.

**Notations.** $\mathbb{R}$ represents the set of real numbers, $\mathbb{Q}$ is the space of quaternion numbers and $\mathbb{Q}^n$ denotes all $n$-dimensional quaternions. $\vartheta = \vartheta^R + \vartheta^I i + \vartheta^J j + \vartheta^K \hbar (\vartheta^R, \vartheta^I, \vartheta^J, \vartheta^K \in \mathbb{R})$

denotes a quaternion number, in which $i, j, \hbar$ are imaginary units of quaternions. The conjugate of quaternion $\vartheta$ is represented by $\vartheta^* = \vartheta^R - \vartheta^I i - \vartheta^J j - \vartheta^K \hbar$, and the sign function is denoted as $\text{sign}(\vartheta) = \text{sign}(\vartheta^R) + \text{sign}(\vartheta^I)i + \text{sign}(\vartheta^J)j + \text{sign}(\vartheta^K)\hbar$. For $\vartheta \in \mathbb{Q}$ and $\epsilon = (\epsilon_1, \epsilon_2, \cdots, \epsilon_n)^T \in \mathbb{Q}^n$, denote $|\vartheta|_1 = \sum_{\varrho=R,I,J,K} |\vartheta^\varrho|$, $\|\epsilon\|_1 = \sum_{i=1}^n |\epsilon_i|_1$, $|\vartheta|_2 = (\sum_{\varrho=R,I,J,K} |\vartheta^\varrho|^2)^{\frac{1}{2}}$, $\|\epsilon\|_2 = (\sum_{i=1}^n |\epsilon_i|_2^2)^{\frac{1}{2}}$, and $\|\epsilon\|_\infty = \max_i\{|\epsilon_i|_1\}$. $C([-\overline{\tau}, 0], \mathbb{Q})$ denotes a continuous mapping from $[-\overline{\tau}, 0]$ to $\mathbb{Q}$, $D^+V(\epsilon(t))$ represents the upper right Dini derivative of $V(\epsilon(t))$.

## II. PRELIMINARIES

Consider the following QVMNN, which incorporates both discrete and distributed time delays:

$$\dot{p}_r(t) = - a_r p_r(t) + \sum_{l=1}^n b_{rl}(p_l(t))f_l(p_l(t))$$
$$+ \sum_{l=1}^n c_{rl}(p_l(t))f_l(p_l(t - \tau(t)))$$
$$+ \sum_{l=1}^n d_{rl}(p_l(t)) \int_{t-\tau}^t f_l(p_l(s))ds, \quad (1)$$

$r, l = 1, 2, \cdots, n$, where $p_r(t) \in \mathbb{Q}$ represents the state variable of the neuron, $a_r > 0$ is the self-inhibition, $f_l(p_l(\cdot)) \in \mathbb{Q}$ is the activation function, $\tau$ and $\tau(t)$ represent the bounded distributed and discrete time delay, respectively. Denote $\overline{\tau} = \max\{\tau, \tau(t)\}$, and the initial condition $p_r(s) = \varphi_r(s) \in C([-\overline{\tau}, 0], \mathbb{Q})$. $b_{rl}(p_l(t))$, $c_{rl}(p_l(t))$ and $d_{rl}(p_l(t))$ are memristors synaptic connection weights which satisfy

$$b_{rl}(p_l(t)) = \begin{cases} \hat{b}_{rl}, & |p_l(t)|_1 \le \kappa_l, \\ \check{b}_{rl}, & |p_l(t)|_1 > \kappa_l, \end{cases}$$

$$c_{rl}(p_l(t)) = \begin{cases} \hat{c}_{rl}, & |p_l(t)|_1 \le \kappa_l, \\ \check{c}_{rl}, & |p_l(t)|_1 > \kappa_l, \end{cases}$$

$$d_{rl}(p_l(t)) = \begin{cases} \hat{d}_{rl}, & |p_l(t)|_1 \le \kappa_l, \\ \check{d}_{rl}, & |p_l(t)|_1 > \kappa_l, \end{cases}$$

where $\hat{b}_{rl}, \check{b}_{rl}, \hat{c}_{rl}, \check{c}_{rl}, \hat{d}_{rl}, \check{d}_{rl} \in \mathbb{Q}$, $\kappa_l$ are the switching jump constants. For $k = 1, 2$, denote $|\tilde{b}_{rl}|_k = \max\{|\hat{b}_{rl}|_k, |\check{b}_{rl}|_k\}$, $|\tilde{c}_{rl}|_k = \max\{|\hat{c}_{rl}|_k, |\check{c}_{rl}|_k\}$, $|\tilde{d}_{rl}|_k = \max\{|\hat{d}_{rl}|_k, |\check{d}_{rl}|_k\}$.

The corresponding response QVMNN can be described by

$$\dot{q}_r(t) = - a_r q_r(t) + \sum_{l=1}^n b_{rl}(q_l(t))f_l(q_l(t))$$
$$+ \sum_{l=1}^n c_{rl}(q_l(t))f_l(q_l(t - \tau(t)))$$
$$+ \sum_{l=1}^n d_{rl}(q_l(t)) \int_{t-\tau}^t f_l(q_l(s))ds + u_r(t), \quad (2)$$

where $q_r(t) \in \mathbb{Q}$ denotes the state variable, $u_r(t)$ is a controller to be designed later. Denote the initial condition of the response system as $q_r(s) = \psi_r(s) \in C([-\overline{\tau}, 0], \mathbb{Q})$.

Now, identify the synchronization error as $\epsilon_r(t) = q_r(t) - p_r(t)$. Let $\epsilon(t) = (\epsilon_1(t), \epsilon_2(t), \cdots, \epsilon_n(t))^T$, the error system can be given by

$$
\begin{aligned}
\dot{\epsilon}_r(t) = & -a_r\epsilon_r(t) + \sum_{l=1}^{n} b_{rl}(q_l(t))g_l(\epsilon_l(t)) + \sum_{l=1}^{n} \big(b_{rl}(q_l(t)) \\
& - b_{rl}(p_l(t))\big)f_l(p_l(t)) + \sum_{l=1}^{n} c_{rl}(q_l(t))g_l(\epsilon_l(t - \tau(t))) \\
& + \sum_{l=1}^{n} \big(c_{rl}(q_l(t)) - c_{rl}(p_l(t))\big)f_l(p_l(t - \tau(t))) \\
& + \sum_{l=1}^{n} d_{rl}(q_l(t)) \int_{t-\tau}^{t} g_l(\epsilon_l(s))ds + \sum_{l=1}^{n} \big(d_{rl}(q_l(t)) \\
& - d_{rl}(p_l(t))\big) \int_{t-\tau}^{t} f_l(p_l(s))ds + u_r(t),
\end{aligned} \tag{3}
$$

where $g_l(\epsilon_l(t)) = f_l(q_l(t)) - f_l(p_l(t))$, denote the initial condition as $\epsilon_r(s) = \psi_r(s) - \varphi_r(s) \in C([-\overline{\tau}, 0], \mathbb{Q})$. Design the controller as

$$
u_r(t) = -\gamma_r\epsilon_r(t) - \omega_r\text{sign}(\epsilon_r(t)), \tag{4}
$$

in which $\gamma_r$ and $\omega_r$ are positive constants. To streamline the analysis of FTS for QVMNNs, certain essential assumptions are proposed.

**Assumption 1.** For $k = 1, 2$, and $l = 1, 2, \cdots, n$, there exist constants $\eta_l > 0$ fulfilling

$$
|f_l(q_l(t)) - f_l(p_l(t))|_k \leq \eta_l|q_l(t) - p_l(t)|_k.
$$

**Assumption 2.** For $k = 1, 2$, and $l = 1, 2, \cdots, n$, $|f_l(q_l(t))|_k$ is bounded. That is, there exist constants $m_l > 0$ fulfilling

$$
|f_l(q_l(t))|_k \leq m_l.
$$

**Definition 1.** [42] The error system (3) is called to be globally stable to $\varpi$ ($\varpi \geq 0$) finite-timely if there exists $T(\epsilon(0)) \geq 0$ fulfilling $\lim_{t \to T(\epsilon(0))} \|\epsilon(t)\| = \varpi$, and for all $t \geq T(\epsilon(0))$, $\|\epsilon(t)\| \leq \varpi$. Especially, system (3) is globally stable to origin finite-timely if $\varpi = 0$.

**Lemma 1.** [42] For $k = 1, 2$, $u, v \in \mathbb{Q}$, these formulas are satisfied.

$$
\begin{aligned}
&(1) u^*\text{sign}(u) + \text{sign}^*(u)u = 2|u|_1, \\
&(2) \text{sign}^*(u)\text{sign}(u) = |\text{sign}(u)|_1, \\
&(3) |uv|_k \leq |u|_k \cdot |v|_k, \\
&(4) u^*v + v^*u \leq 2|u|_k \cdot |v|_k, |u|_2 \leq |u|_1.
\end{aligned}
$$

**Remark 1.** By introducing these inequalities, the FTS of QVMNNs is analyzed by non-decomposition method, which help to avoid complex operations involved in the decomposition method and enriches the theoretical framework of non-decomposition methods for QVNNs.

**Lemma 2.** [43] For $V(\epsilon(t)) : [t_0 - \overline{\tau}, +\infty) \to [0, +\infty)$, if there exist $\alpha > \beta > 0$ such that

$$
D^+V(\epsilon(t)) \leq -\alpha V(\epsilon(t)) + \beta\tilde{V}(\epsilon(t)),
$$

then for $t \geq t_0$, the following inequality hold

$$
V(\epsilon(t)) \leq \tilde{V}(\epsilon(t_0))e^{-\delta(t-t_0)},
$$

where $\tilde{V}(\epsilon(t)) = \sup_{t-\overline{\tau} \leq s \leq t} V(\epsilon(s))$, $\delta > 0$ and satisfies $\delta = \alpha - \beta e^{\delta\overline{\tau}}$.

**Lemma 3.** [44] The response QVMNN (2) is finite-time synchronized with QVMNN (1), if for error system (3), there exists a regular, positive definite and radially unbounded function $V(\epsilon(t))$ satisfying

$$
D^+V(\epsilon(t)) \leq -\zeta V^{\mu}(\epsilon(t)) - \xi,
$$

where $\zeta > 0$, $\xi \geq 0$, and $0 < \mu \leq 1$. Specially,
(1) if $\xi = 0$ and $0 < \mu < 1$, the synchronization time can be inferred by

$$
T(\epsilon(0)) \leq \frac{V^{1-\mu}(\epsilon(0))}{\zeta(1-\mu)};
$$

(2) if $\xi > 0$ and $\mu = 1$, the synchronization time can be inferred by

$$
T(\epsilon(0)) \leq \frac{1}{\zeta} \ln \big(1 + \frac{\zeta}{\xi}V(\epsilon(0))\big).
$$

## III. MAIN RESULTS

This section presents a set of sufficient criteria for FTS control of QVMNNs, by constructing Lyapunov functions by distinct norm forms. The analysis of FTS entails the error system converging from its initial state to the unit ball within a finite time, followed by subsequent convergence from the unit ball to the origin finite-timely. For conciseness, denote

$$
\begin{aligned}
F_{r,k} &= \sum_{l=1}^{n} \eta_l \big(|\tilde{b}_{rl}|_k + |\tilde{c}_{rl}|_k + \tau|\tilde{d}_{rl}|_k\big), \\
\Delta_{r,k} &= \sum_{l=1}^{n} m_l \big(|\hat{b}_{rl} - \check{b}_{rl}|_k + |\hat{c}_{rl} - \check{c}_{rl}|_k + \tau|\hat{d}_{rl} - \check{d}_{rl}|_k\big), \\
\Xi_{r,k} &= \sum_{l=1}^{n} \eta_l \big(|\tilde{c}_{rl}|_k + \tau|\tilde{d}_{rl}|_k\big), \\
\beth_{r,k} &= \eta_r \sum_{l=1}^{n} |\tilde{b}_{lr}|_k,
\end{aligned}
$$

where $r = 1, 2, \cdots, n$, $k = 1, 2$.

**Theorem 1.** Suppose that Assumptions 1 and 2 hold, the response QVMNN (2) can be synchronized with QVMNN (1) within a finite time, if the controller (4) fulfills the following inequalities

$$
\gamma_r + a_r > \max\left\{\frac{\eta}{2} \sum_{r=1}^{n} \big(|\tilde{c}_{r\cdot}|_2 + \tau|\tilde{d}_{r\cdot}|_2\big) + \frac{1}{2}\big(F_{r,2} + \beth_{r,2}\big),\right.
$$
$$
\left.\frac{1}{2} \sum_{l=1}^{n} \big(\eta_l|\tilde{b}_{rl}|_2 + \eta_r|\tilde{b}_{lr}|_2\big)\right\}, \tag{5}
$$

$$
\omega_r \geq \Delta_{r,2} + \Xi_{r,2} + \varepsilon, \tag{6}
$$

where $\varepsilon > 0$, $|\tilde{c}_{r\cdot}|_2 = \max_l\{|\tilde{c}_{rl}|_2\}$, $|\tilde{d}_{r\cdot}|_2 = \max_l\{|\tilde{d}_{rl}|_2\}$, $\eta = \max_r\{\eta_r\}$. Moreover, it can be achieved within time

$$T = \begin{cases} \dfrac{1}{\varepsilon}\|\epsilon(0)\|_2, & \text{if } \sup_{-\overline{\tau}\leq s\leq 0}\|\epsilon(s)\|_2 \leq 1, \\[2mm] \dfrac{1}{\delta}\ln\left(\sup_{-\overline{\tau}\leq s\leq 0}\|\epsilon(s)\|_2\right) + \overline{\tau} + \dfrac{1}{\varepsilon}, & \text{otherwise,} \end{cases}$$

where $\delta$ can be solved by $\delta = \alpha - \beta e^{\delta\overline{\tau}}$, $\alpha_r = 2(a_r + \gamma_r) - F_{r,2} - \beth_{r,2}$, $\alpha = \min_r\{\alpha_r\}$, and $\beta = \eta\sum_{r=1}^{n}(|\tilde{c}_{r\cdot}|_2 + \tau|\tilde{d}_{r\cdot}|_2)$.

**Proof.** Construct the Lyapunov function as

$$V(\epsilon(t)) = \frac{1}{2}\sum_{r=1}^{n}\epsilon_r^*(t)\epsilon_r(t).$$

Analysing the Dini derivative of $V(\epsilon(t))$ yields that

$$D^+V(\epsilon(t))$$
$$= \frac{1}{2}\sum_{r=1}^{n}\epsilon_r^*(t)\dot{\epsilon}_r(t) + \dot{\epsilon}_r^*(t)\epsilon_r(t)$$
$$= \frac{1}{2}\sum_{r=1}^{n}\left(\epsilon_r^*(t)(-a_r)\epsilon_r(t) + \epsilon_r^*(t)(-a_r)^*\epsilon_r(t)\right)$$
$$+ \frac{1}{2}\sum_{r=1}^{n}\left(\epsilon_r^*(t)\sum_{l=1}^{n}b_{rl}(q_l(t))g_l(\epsilon_l(t)) + \left(\sum_{l=1}^{n}b_{rl}(q_l(t))\right.\right.$$
$$\left.\times g_l(\epsilon_l(t))\right)^*\epsilon_r(t)\right) + \frac{1}{2}\sum_{r=1}^{n}\left(\epsilon_r^*(t)\sum_{l=1}^{n}\left(b_{rl}(q_l(t))\right.\right.$$
$$- b_{rl}(p_l(t)))f_l(p_l(t)) + \left(\sum_{l=1}^{n}(b_{rl}(q_l(t)) - b_{rl}(p_l(t)))\right.$$
$$\left.\left.\times f_l(p_l(t))\right)^*\epsilon_r(t)\right) + \frac{1}{2}\sum_{r=1}^{n}\left(\epsilon_r^*(t)\sum_{l=1}^{n}c_{rl}(q_l(t))\right.$$
$$\times g_l(\epsilon_l(t - \tau(t))) + \left(\sum_{l=1}^{n}c_{rl}(q_l(t))g_l(\epsilon_l(t - \tau(t)))\right)^*$$
$$\left.\times \epsilon_r(t)\right) + \frac{1}{2}\sum_{r=1}^{n}\left(\epsilon_r^*(t)\sum_{l=1}^{n}(c_{rl}(q_l(t)) - c_{rl}(p_l(t)))\right.$$
$$\times f_l(p_l(t - \tau(t))) + \left(\sum_{l=1}^{n}(c_{rl}(q_l(t)) - c_{rl}(p_l(t)))\right.$$
$$\left.\left.\times f_l(p_l(t - \tau(t)))\right)^*\epsilon_r(t)\right) + \frac{1}{2}\sum_{r=1}^{n}\left(\epsilon_r^*(t)\sum_{l=1}^{n}d_{rl}(q_l(t))\right.$$
$$\times \int_{t-\tau}^{t}g_l(\epsilon_l(s))ds + \left(\sum_{l=1}^{n}d_{rl}(q_l(t))\int_{t-\tau}^{t}g_l(\epsilon_l(s))ds\right)^*$$
$$\left.\times \epsilon_r(t)\right) + \frac{1}{2}\sum_{r=1}^{n}\left(\epsilon_r^*(t)\sum_{l=1}^{n}(d_{rl}(q_l(t)) - d_{rl}(p_l(t)))\right.$$
$$\times \int_{t-\tau}^{t}f_l(p_l(s))ds + \left(\sum_{l=1}^{n}(d_{rl}(q_l(t)) - d_{rl}(p_l(t)))\right.$$
$$\left.\left.\times \int_{t-\tau}^{t}f_l(p_l(s))ds\right)^*\epsilon_l(t)\right) + \frac{1}{2}\sum_{r=1}^{n}\left(\epsilon_r^*(t)(-\gamma_r)\right.$$

$$\times \epsilon_r(t) + \epsilon_r^*(t)(-\gamma_r)^*\epsilon_r(t)) + \frac{1}{2}\sum_{r=1}^{n}\left(\epsilon_r^*(t)(-\omega_r)\right.$$
$$\times \text{sign}(\epsilon_r(t)) + \text{sign}^*(\epsilon_r(t))(-\omega_r)^*\epsilon_r(t)).$$

From Lemma 1, it is obvious to see

$$\frac{1}{2}\sum_{r=1}^{n}\left(\epsilon_r^*(t)(-a_r)\epsilon_r(t) + \epsilon_r^*(t)(-a_r)^*\epsilon_r(t)\right)$$
$$+ \frac{1}{2}\sum_{r=1}^{n}\left(\epsilon_r^*(t)(-\gamma_r)\epsilon_r(t) + \epsilon_r^*(t)(-\gamma_r)^*\epsilon_r(t)\right)$$
$$= -\sum_{r=1}^{n}(a_r + \gamma_r)\epsilon_r^*(t)\epsilon_r(t).$$

Besides, under the Assumption 1 and Assumption 2, we have

$$\frac{1}{2}\sum_{r=1}^{n}\left(\epsilon_r^*(t)\sum_{l=1}^{n}b_{rl}(q_l(t))g_l(\epsilon_l(t))\right.$$
$$+ \left(\sum_{l=1}^{n}b_{rl}(q_l(t))g_l(\epsilon_l(t))\right)^*\epsilon_r(t)\right)$$
$$\leq \frac{1}{2}\sum_{r=1}^{n}\sum_{l=1}^{n}|\tilde{b}_{rl}|_2\eta_l\left(|\epsilon_r(t)|_2^2 + |\epsilon_l(t)|_2^2\right)$$
$$= \frac{1}{2}\sum_{r=1}^{n}\sum_{l=1}^{n}\left(\eta_l|\tilde{b}_{rl}|_2 + \eta_r|\tilde{b}_{lr}|_2\right)|\epsilon_r(t)|_2^2,$$

and

$$\frac{1}{2}\sum_{r=1}^{n}\left(\epsilon_r^*(t)\sum_{l=1}^{n}c_{rl}(q_l(t))g_l(\epsilon_l(t - \tau(t)))\right.$$
$$+ \left(\sum_{l=1}^{n}c_{rl}(q_l(t))g_l(\epsilon_l(t - \tau(t)))\right)^*\epsilon_r(t)\right)$$
$$+ \frac{1}{2}\sum_{r=1}^{n}\left(\epsilon_r^*(t)\sum_{l=1}^{n}d_{rl}(q_l(t))\int_{t-\tau}^{t}g_l(\epsilon_l(s))ds\right.$$
$$+ \left(\sum_{l=1}^{n}d_{rl}(q_l(t))\int_{t-\tau}^{t}g_l(\epsilon_l(s))ds\right)^*\epsilon_r(t)\right)$$
$$\leq \sum_{r=1}^{n}|\epsilon_r(t)|_2\sum_{l=1}^{n}|\tilde{c}_{rl}|_2\eta_l|\epsilon_l(t - \tau(t))|_2$$
$$+ \sum_{r=1}^{n}|\epsilon_r(t)|_2\sum_{l=1}^{n}|\tilde{d}_{rl}|_2\int_{t-\tau}^{t}\eta_l|\epsilon_l(s)|_2ds$$
$$\leq \frac{1}{2}\sum_{r=1}^{n}\sum_{l=1}^{n}\eta_l\left(|\tilde{c}_{rl}|_2 + \tau|\tilde{d}_{rl}|_2\right)|\epsilon_r(t)|_2^2$$
$$+ \frac{1}{2}\eta\sum_{r=1}^{n}\left(|\tilde{c}_{r\cdot}|_2 + \tau|\tilde{d}_{r\cdot}|_2\right)\sup_{t-\overline{\tau}\leq s\leq t}\|\epsilon(s)\|_2^2.$$

Then, combining with the above analysis yields that

$$D^+V(\epsilon(t))$$
$$\leq -\sum_{r=1}^{n}\left(a_r + \gamma_r\right)|\epsilon_r(t)|_2^2 + \sum_{r=1}^{n}\sum_{l=1}^{n}m_l\left(|\hat{b}_{rl} - \check{b}_{rl}|_2\right.$$
$$+ |\hat{c}_{rl} - \check{c}_{rl}|_2 + \tau|\hat{d}_{rl} - \check{d}_{rl}|_2\right)|\epsilon_r(t)|_2$$

$$+ \frac{1}{2} \sum_{r=1}^{n} \sum_{l=1}^{n} \Big( \eta_l |\tilde{b}_{rl}|_2 + \eta_r |\tilde{b}_{lr}|_2 \Big) |\epsilon_r(t)|_2^2$$

$$+ \frac{1}{2} \sum_{r=1}^{n} \sum_{l=1}^{n} \eta_l \Big( |\tilde{c}_{rl}|_2 + \tau |\tilde{d}_{rl}|_2 \Big) |\epsilon_r(t)|_2^2$$

$$+ \frac{1}{2} \eta \sum_{r=1}^{n} \Big( |\tilde{c}_{r\cdot}|_2 + \tau |\tilde{d}_{r\cdot}|_2 \Big) \sup_{t - \overline{\tau} \le s \le t} \|\epsilon(s)\|_2^2$$

$$- \sum_{r=1}^{n} \omega_r |\epsilon_r(t)|_2$$

$$\le - \sum_{r=1}^{n} \Big( a_r + \gamma_r - \frac{1}{2} F_{r,2} - \frac{1}{2} \beth_{r,2} \Big) |\epsilon_r(t)|_2^2$$

$$+ \frac{1}{2} \eta \sum_{r=1}^{n} \Big( |\tilde{c}_{r\cdot}|_2 + \tau |\tilde{d}_{r\cdot}|_2 \Big) \sup_{t - \overline{\tau} \le s \le t} \|\epsilon(s)\|_2^2$$

$$- \sum_{r=1}^{n} \Big( \omega_r - \Delta_{r,2} \Big) |\epsilon_r(t)|_2.$$

Denote $\alpha_r = 2(a_r + \gamma_r) - F_{r,2} - \beth_{r,2}$, $\alpha = \min_r \{\alpha_r\}$, and $\beta = \eta \sum_{r=1}^{n} (|\tilde{c}_{r\cdot}|_2 + \tau |\tilde{d}_{r\cdot}|_2)$. Suppose the initial condition satisfies $\sup_{-\overline{\tau} \le s \le 0} \|\epsilon(s)\|_2 > 1$, then under the conditions (5) and (6), we have

$$D^+ V(\epsilon(t)) \le - \alpha V(\epsilon(t)) + \beta \tilde{V}(\epsilon(t)), \qquad (7)$$

where $\tilde{V}(\epsilon(t)) = \sup_{t - \overline{\tau} \le s \le t} V(\epsilon(s))$. Based on Lemma 2, $V(\epsilon(t))$ satisfies the inequality $V(\epsilon(t)) \le \tilde{V}(\epsilon(0)) e^{-\delta t}$, where $\delta = \alpha - \beta e^{\delta \overline{\tau}}$. Furthermore, it is obvious from (7) that there must exist an instant $t_1 \le \frac{1}{\delta} \ln(2\tilde{V}(\epsilon(0)))$ such that $\|\epsilon(s)\|_2 \le 1$ when $t_1 \le s \le t_1 + \overline{\tau}$. Thus, the error system (3) can be stable to 1 within time

$$T_1 \le \frac{1}{\delta} \ln \big( 2\tilde{V}(\epsilon(0)) \big) + \overline{\tau}.$$

Next, it can be proved that under the conditions (5) and (6), $\sup_{t - \overline{\tau} \le s \le t} \|\epsilon(s)\|_2 < 1$ is fulfilled after $T_1$. Combined with $\|\epsilon(t)\|_2 < 1, t \in (T_1 - \overline{\tau}, T_1)$, it follows that $\|\epsilon(t)\|_2 < 1$ is satisfied for $t > T_1$. Otherwise, there exists an instant $t' = \inf\{t > T_1 | \|\epsilon(t)\|_2 = 1\}$, and there is $T_1 \le \hat{t} < t' < \infty$, such that $D^+ \|\epsilon(t)\|_2 > 0$, for $t \in (\hat{t}, t')$. On the other hand, as condition $\sup_{t - \overline{\tau} \le s \le t} \|\epsilon(s)\|_2 < 1, t \in (\hat{t}, t')$ holds, by computing the Dini derivative of $V(\epsilon(t))$, we derive that

$$D^+ V(\epsilon(t))$$

$$\le - \sum_{r=1}^{n} \big( a_r + \gamma_r \big) |\epsilon_r(t)|_2^2 + \frac{1}{2} \sum_{r=1}^{n} \sum_{l=1}^{n} \big( \eta_l |\tilde{b}_{rl}|_2 + \eta_r |\tilde{b}_{lr}|_2 \big)$$

$$\times |\epsilon_r(t)|_2^2 - \sum_{r=1}^{n} \Big( \omega_r - \sum_{l=1}^{n} m_l (|\hat{b}_{rl} - \check{b}_{rl}|_2 + |\hat{c}_{rl} - \check{c}_{rl}|_2$$

$$+ \tau |\hat{d}_{rl} - \check{d}_{rl}|_2) \Big) |\epsilon_r(t)|_2 + \sum_{r=1}^{n} \sum_{l=1}^{n} |\tilde{c}_{rl}|_2 \eta_l |\epsilon_l(t - \tau(t))|_2$$

$$\times |\epsilon_r(t)|_2 + \sum_{r=1}^{n} \sum_{l=1}^{n} |\tilde{d}_{rl}|_2 \tau \eta_l |\epsilon_r(t)|_2 \sup_{t - \overline{\tau} \le s \le t} |\epsilon_l(s)|_2$$

$$\le - \sum_{r=1}^{n} \Big( a_r + \gamma_r - \frac{1}{2} \sum_{l=1}^{n} \big( \eta_l |\tilde{b}_{rl}|_2 + \eta_r |\tilde{b}_{lr}|_2 \big) \Big) |\epsilon_r(t)|_2^2$$

$$- \sum_{r=1}^{n} \Big( \omega_r - \Delta_{r,2} - \Xi_{r,2} \Big) |\epsilon_r(t)|_2.$$

It can be derived from the conditions (5) and (6) that

$$D^+ V(\epsilon(t)) \le - \varepsilon \sum_{r=1}^{n} |\epsilon_r(t)|_2 \le -\sqrt{2}\varepsilon V^{\frac{1}{2}}(\epsilon(t)),$$

it contradicts the hypothesis. Thus, $\sup_{t - \overline{\tau} \le s \le t} \|\epsilon(s)\|_2 < 1$ is always satisfied after $T_1$. Furthermore, from the above analysis and Lemma 3, the error system (3) can be stable from 1 to 0 within time

$$T_2 \le \frac{2V^{\frac{1}{2}}(\epsilon(T_1))}{\sqrt{2}\varepsilon} \le \frac{1}{\varepsilon}.$$

Thus, if the initial condition $\sup_{-\overline{\tau} \le s \le 0} \|\epsilon(s)\|_2 < 1$, the response QVMNN (2) can synchronize with QVMNN (1) within a period of no more than $T_2$, to be more specific, it can be estimated by $\frac{\|\epsilon(0)\|_2}{\varepsilon}$. If the initial value is larger than 1, the total synchronization time is estimated by $T = T_1 + T_2$.

**Remark 2.** Obviously, this paper constructs Lyapunov functions in simple norm forms, instead of the complicated Lyapunov functionals constructed in [35], [36], and [42]. Notably, the constraint $\dot{\tau}(t) < 1$ is eliminated.

Theorem 1 establishes criteria for achieving FTS of considered QVMNNs under the simple controller (4). Specifically, the Lyapunov function is formed by 2-norm. Subsequently, this approach is proved to be also effective in 1-norm and $\infty$-norm forms.

**Theorem 2.** Assuming that Assumptions 1 and 2 are met, QVMNNs (1) and (2) can be synchronized within a finite time, if the controller (4) satisfies the following inequalities

$$\gamma_r + a_r > \beth_{r,1} + \eta \sum_{r=1}^{n} \big( |\tilde{c}_{r\cdot}|_1 + \tau |\tilde{d}_{r\cdot}|_1 \big), \qquad (8)$$

$$\omega_r \ge \sum_{r=1}^{n} \Big( \Delta_{r,1} + \Xi_{r,1} \Big) + \xi, \qquad (9)$$

where $\xi$ is a positive constant. Moreover, it can be achieved within time

$$T = \begin{cases} \dfrac{1}{\zeta} \ln \Big( 1 + \dfrac{\zeta}{\xi} \|\epsilon(0)\|_1 \Big), & \text{if} \quad \sup_{-\overline{\tau} \le s \le 0} \|\epsilon(s)\|_1 \le 1, \\[2ex] \dfrac{1}{\delta} \ln \sup_{-\overline{\tau} \le s \le 0} \|\epsilon(s)\|_1 + \overline{\tau} + \dfrac{1}{\zeta} \ln \Big( 1 + \dfrac{\zeta}{\xi} \Big), & \text{otherwise,} \end{cases}$$

where $\zeta = \eta \sum_{r=1}^{n} \big( |\tilde{c}_{r\cdot}|_1 + \tau |\tilde{d}_{r\cdot}|_1 \big)$, $\delta = \alpha - \beta e^{\delta \overline{\tau}}$, $\alpha = \min_r \{\alpha_r\}$, $\alpha_r = a_r + \gamma_r - \beth_{r,1}$, and $\beta = \eta \sum_{r=1}^{n} \big( |\tilde{c}_{r\cdot}|_1 + \tau |\tilde{d}_{r\cdot}|_1 \big)$.

**Proof.** Applying the 1-norm of quaternions, consider the Lyapunov function as

$$W(\epsilon(t)) = \frac{1}{2} \sum_{r=1}^{n} \Big( \text{sign}^*(\epsilon_r(t)) \epsilon_r(t) + \epsilon_r^*(t) \text{sign}(\epsilon_r(t)) \Big).$$

Supposed the initial condition fulfills $\sup_{-\overline{\tau} \le s \le 0} \|\epsilon(s)\|_1 \ge 1$, then the Dini derivative $W(\epsilon(t))$ is calculated as

$$D^+ W(\epsilon(t))$$

$$= \frac{1}{2} \sum_{r=1}^{n} \left( \text{sign}^*(\epsilon_r(t)) \dot{\epsilon}_r(t) + \dot{\epsilon}_r^*(t) \text{sign}(\epsilon_r(t)) \right)$$

$$\le -\sum_{r=1}^{n} a_r |\epsilon_r(t)|_1 + \sum_{r=1}^{n} \sum_{l=1}^{n} m_l \left( |\hat{b}_{rl} - \check{b}_{rl}|_1 + |\hat{c}_{rl} - \check{c}_{rl}|_1 \right.$$

$$+ \tau |\hat{d}_{rl} - \check{d}_{rl}|_1 \Big) + \sum_{r=1}^{n} \sum_{l=1}^{n} |\tilde{b}_{rl}|_1 \eta_l |\epsilon_l(t)|_1 + \sum_{r=1}^{n} \sum_{l=1}^{n} |\tilde{c}_{rl}|_1$$

$$\times \eta_l |\epsilon_l(t - \tau(t))|_1 + \sum_{r=1}^{n} \sum_{l=1}^{n} |\tilde{d}_{rl}|_1 \int_{t-\tau}^{t} \eta_l |\epsilon_l(s)|_1 ds$$

$$- \sum_{r=1}^{n} \gamma_r |\epsilon_r(t)|_1 - \omega_{\min} \sum_{r=1}^{n} |\text{sign}(\epsilon_r(t))|_1$$

$$\le -\sum_{r=1}^{n} \left( a_r + \gamma_r - \beth_{r,1} \right) |\epsilon_r(t)|_1$$

$$+ \eta \sum_{r=1}^{n} \left( |\tilde{c}_{r\cdot}|_1 + \tau |\tilde{d}_{r\cdot}|_1 \right) \sup_{t-\overline{\tau} \le s \le t} \|\epsilon(s)\|_1$$

$$- \left( \omega_{\min} - \sum_{r=1}^{n} \Delta_{r,1} \right),$$

where $\omega_{\min} = \min_r \{\omega_r\}$. Let $\alpha = \min_r \{\alpha_r\}$, $\alpha_r = a_r + \gamma_r - \beth_{r,1}$, and $\beta = \eta \sum_{r=1}^{n} \left( |\tilde{c}_{r\cdot}|_1 + \tau |\tilde{d}_{r\cdot}|_1 \right)$. Then the following inequality can be obtained under conditions (8) and (9)

$$D^+ W(\epsilon(t)) \le -\alpha W(\epsilon(t)) + \beta \tilde{W}(\epsilon(t)),$$

where $\tilde{W}(\epsilon(0)) = \sup_{-\overline{\tau} \le s \le 0} W(\epsilon(s))$. From Lemma 2, it has $W(\epsilon(t)) \le \tilde{W}(\epsilon(0)) e^{-\delta t}$, where $\delta = \alpha - \beta e^{\delta \overline{\tau}}$. Similar to the analysis in Theorem 1, there must exist an instant $t_1 \le \frac{1}{\delta} \ln \tilde{W}(\epsilon(0))$, such that $W(\epsilon(s)) \le 1$ when $s \in [t_1, t_1 + \overline{\tau}]$. Thus, if the initial value is larger than 1, the error system (3) can be stable to 1 within time $T_1 \le \frac{1}{\delta} \ln \tilde{W}(\epsilon(0)) + \overline{\tau}$.

Similarly, $\sup_{t-\overline{\tau} \le s \le t} \|\epsilon(s)\|_1 < 1$ always holds after $T_1$, and under the conditions (8) and (9), the Dini derivative of $W(\epsilon(t))$ satisfies

$$D^+ W(\epsilon(t))$$

$$\le -\sum_{r=1}^{n} (a_r + \gamma_r) |\epsilon_r(t)|_1 + \sum_{r=1}^{n} \sum_{l=1}^{n} |\tilde{b}_{lr}|_1 \eta_r |\epsilon_r(t)|_1$$

$$+ \sum_{r=1}^{n} \sum_{l=1}^{n} m_l \left( |\hat{b}_{rl} - \check{b}_{rl}|_1 + |\hat{c}_{rl} - \check{c}_{rl}|_1 + \tau |\hat{d}_{rl} - \check{d}_{rl}|_1 \right)$$

$$+ \sum_{r=1}^{n} \sum_{l=1}^{n} \eta_l |\tilde{c}_{rl}|_1 + \sum_{r=1}^{n} \sum_{l=1}^{n} |\tilde{d}_{rl}|_1 \tau \eta_l \sup_{t-\overline{\tau} \le s \le t} |\epsilon_l(s)|_1$$

$$- \omega_{\min} \sum_{r=1}^{n} |\text{sign}(\epsilon_r(t))|_1$$

$$\le -\sum_{r=1}^{n} \left( a_r + \gamma_r - \beth_{r,1} \right) |\epsilon_r(t)|_1$$

$$- \omega_{\min} + \sum_{r=1}^{n} \left( \Delta_{r,1} + \Xi_{r,1} \right)$$

$$\le -\zeta W(\epsilon(t)) - \xi.$$

By Lemma 3, it can be concluded that after time $T_1$, the error system (3) can globally stable to origin within time

$$T_2 \le \frac{1}{\zeta} \ln \left( 1 + \frac{\zeta}{\xi} W(\epsilon(T_1)) \right) \le \frac{1}{\zeta} \ln(1 + \frac{\zeta}{\xi}).$$

Consequently, if $\sup_{-\overline{\tau} \le s \le 0} \|\epsilon(s)\|_1 > 1$, the drive-response QVMNNs can realize FTS within $T = T_1 + T_2$. Otherwise, the synchronization can be realized within $\frac{1}{\zeta} \ln(1 + \frac{\zeta}{\xi} \|\epsilon(0)\|_1)$.

Incorporating the improved $\infty$-norm enables to extend the previous conclusions and arrive at the subsequent outcome.

**Theorem 3.** Supposing Assumptions 1 and 2 are fulfilled, the response QVMNN (2) can be synchronized with QVMNN (1) finite-timely, if the controller (4) fulfills the following inequalities

$$\gamma_r + a_r > \max_r \left\{ \sum_{l=1}^{n} \eta_l (|\tilde{c}_{rl}|_1 + \tau |\tilde{d}_{rl}|_1) \right\} + \sum_{l=1}^{n} \eta_l |\tilde{b}_{rl}|_1,$$
(10)

$$\omega_r \ge \Delta_{r,1} + \Xi_{r,1} + \xi,$$
(11)

where $\xi > 0$. Moreover, it can be achieved within time

$$T = \begin{cases} \dfrac{1}{\zeta} \ln \left( 1 + \dfrac{\zeta}{\xi} \|\epsilon(0)\|_\infty \right), & \text{if } \sup_{-\overline{\tau} \le s \le 0} \|\epsilon(s)\|_\infty \le 1, \\ \dfrac{1}{\delta} \ln \sup_{-\overline{\tau} \le s \le 0} \|\epsilon(s)\|_\infty + \overline{\tau} + \dfrac{1}{\zeta} \ln \left( 1 + \dfrac{\zeta}{\xi} \right), & \text{otherwise,} \end{cases}$$

where $\zeta = \max_r \left\{ \sum_{l=1}^{n} \eta_l (|\tilde{c}_{rl}|_1 + \tau |\tilde{d}_{rl}|_1) \right\}$, $\delta = \alpha - \beta e^{\delta \overline{\tau}}$, $\alpha = \min_i \{\alpha_i\}$, $\alpha_i = a_i + \gamma_i - \sum_{l=1}^{n} \eta_l |\tilde{b}_{il}|_1$, and $\beta = \max_i \{\beta_i\}$, $\beta_i = \sum_{l=1}^{n} \eta_l (|\tilde{c}_{il}|_1 + \tau |\tilde{d}_{il}|_1)$.

**Proof.** Construct the following Lyapunov function

$$Z(\epsilon(t)) = \|\epsilon(t)\|_\infty,$$

obviously, there exists an index $i \in \{1, 2, \cdots, n\}$ such that $Z(\epsilon(t)) = |\epsilon_i(t)|_1$. Similarly, calculating the Dini derivative of $Z(\epsilon(t))$ yields that

$$D^+ Z(\epsilon(t))$$

$$= \frac{1}{2} \left( \text{sign}^*(\epsilon_i(t)) \dot{\epsilon}_i(t) + \dot{\epsilon}_i^*(t) \text{sign}(\epsilon_i(t)) \right)$$

$$\le -a_i |\epsilon_i(t)|_1 + \sum_{l=1}^{n} |\tilde{b}_{il}|_1 \eta_l |\epsilon_l(t)|_1 + \sum_{l=1}^{n} |\hat{b}_{il} - \check{b}_{il}|_1 m_l$$

$$+ \sum_{l=1}^{n} |\tilde{c}_{il}|_1 \eta_l |\epsilon_l(t - \tau(t))|_1 + \sum_{l=1}^{n} |\hat{c}_{il} - \check{c}_{il}|_1 m_l$$

$$+ \sum_{l=1}^{n} \tau |\tilde{d}_{il}|_1 \eta_l \sup_{t-\overline{\tau} \le s \le t} |\epsilon_l(s)|_1 + \sum_{l=1}^{n} \tau |\hat{d}_{il} - \check{d}_{il}|_1 m_l$$

$$- \gamma_i |\epsilon_i(t)|_1 - \omega_i |\text{sign}(\epsilon_i(t))|_1$$

$$\le - \left( a_i + \gamma_i - \sum_{l=1}^{n} \eta_l |\tilde{b}_{il}|_1 \right) \|\epsilon(t)\|_\infty + \sum_{l=1}^{n} \eta_l \left( |\tilde{c}_{il}|_1 + \tau \right.$$

$$\times |\tilde{d}_{il}|_1) \sup_{t-\overline{\tau} \le s \le t} \|\epsilon(s)\|_\infty - \left(\omega_i - \Delta_{i,1}\right).$$

Similarly, denote $\alpha_i = a_i + \gamma_i - \sum_{l=1}^n \eta_l |\tilde{b}_{il}|_1$, $\alpha = \min_i\{\alpha_i\}$ and $\beta_i = \sum_{l=1}^n \eta_l(|\tilde{c}_{il}|_1 + \tau|\tilde{d}_{il}|_1)$, $\beta = \max_i\{\beta_i\}$. Then, it follows from (10), (11) and Lemma 2 that $Z(\epsilon(t)) \le \tilde{Z}(\epsilon(0))e^{-\delta t}$, where $\delta = \alpha - \beta e^{\delta \overline{\tau}}$. Therefore, there exists an instant $t_1 \le \frac{1}{\delta} \ln \tilde{Z}(\epsilon(0))$, such that $Z(\epsilon(s)) \le 1$ when $s \in [t_1, t_1 + \overline{\tau}]$. So the error system (3) can be stable to 1 within the period $T_1 \le \frac{1}{\delta} \ln \tilde{Z}(\epsilon(0)) + \overline{\tau}$.

After time $T_1$, the Dini derivative of $Z(\epsilon(t))$ fulfills

$$D^+ Z(\epsilon(t))$$
$$\le -\left(a_i + \gamma_i - \sum_{l=1}^n \eta_l |\tilde{b}_{il}|_1\right)\|\epsilon(t)\|_\infty - \omega_i + \Delta_{i,1} + \Xi_{i,1},$$

if conditions (10) and (11) are hold. Then from Lemma 3, the synchronization error can converge to zero within $T_2 \le \frac{1}{\zeta} \ln(1 + \frac{\zeta}{\xi})$, where $\zeta = \max_r \left\{ \sum_{l=1}^n \eta_l(|\tilde{c}_{rl}|_1 + \tau|\tilde{d}_{rl}|_1) \right\}$. Combing the above analysis, if the initial condition satisfies $\sup_{-\overline{\tau} \le s \le 0} \|\epsilon(s)\|_\infty > 1$, the QVMNN (2) can synchronize with QVMNN (1) within $T = T_1 + T_2$.

**Remark 3.** By applying $\infty$-norm in the construction of Lyapunov function, some algebraic criteria are derived to guarantee FTS of QVMNNs. In contrast to existing research that employ non-decomposition methods by 1-norm or 2-norm, this paper explores three distinct quaternion norms, providing a wider selection of control gains for FTS analysis.

**Remark 4.** In some studies for FTS of delayed MNNs, scholars designed controllers which contain the expression for time delays [37], [38]. However, it can be challenging to accurately predict or obtain such information in practical applications. It is worth pointing out that the controller (4) contains only linear and sign function terms. By applying it, Theorems 1-3 provide a set of algebraic criteria for FTS. Compared to the controller in [42], this controller is easier to operate and does not require switching.

## IV. NUMERICAL SIMULATIONS

This section will provide numerical simulations to illustrate the effectiveness of the theoretical results and the application to secure communication.

**Example 1.** Consider the following 2-D drive-response QVMNNs (1) and (2), where $a_1 = a_2 = 1.2$, $f_l(p_l(t)) = 0.5\tanh(p_l^R(t)) + 0.5\tanh(p_l^I(t))i + 0.5\tanh(p_l^J(t))j + 0.5\tanh(p_l^K(t))\hbar$, for $l = 1, 2$, $\tau(t) = |\sin(t)|$, $\tau = 0.1$, $\hat{b}_{22} = \check{b}_{22} = 1.5 - 1.2i + 0.5j + 1.1\hbar$, $\hat{d}_{22} = \check{d}_{22} = 0.2 + 1.3i + j - 1.2\hbar$,

$$b_{11}(p_1(t)) = \begin{cases} 1.7 - 1.3i - 1.6j - 1.1\hbar, & |p_1(t)|_1 \le 2, \\ 1.8 - 1.6i - 1.3j - 1.3\hbar, & |p_1(t)|_1 > 2, \end{cases}$$

$$b_{12}(p_2(t)) = \begin{cases} -0.5 + 1.2i - 1.9j + 1.5\hbar, & |p_2(t)|_1 \le 2, \\ -0.6 + 1.5i - 1.7j + 1.3\hbar, & |p_2(t)|_1 > 2, \end{cases}$$

$$b_{21}(p_1(t)) = \begin{cases} 1.0 + 0.1i - 1.9j + 1.3\hbar, & |p_1(t)|_1 \le 2, \\ 0.8 - 0.1i - 1.7j + 1.3\hbar, & |p_1(t)|_1 > 2, \end{cases}$$

$$c_{11}(p_1(t)) = \begin{cases} -0.3 + 0.4i + 1.6j + 1.3\hbar, & |p_1(t)|_1 \le 2, \\ -0.2 + 0.3i + 2.0j + 1.3\hbar, & |p_1(t)|_1 > 2, \end{cases}$$

$$c_{12}(p_2(t)) = \begin{cases} 1.3 - 0.5i - 0.9j + 1.5\hbar, & |p_2(t)|_1 \le 2, \\ 1.3 - 0.5i - 1.3j + 1.4\hbar, & |p_2(t)|_1 > 2, \end{cases}$$

$$c_{21}(p_1(t)) = \begin{cases} 1.3 - 0.4i - 1.5j + 1.2\hbar, & |p_1(t)|_1 \le 2, \\ 1.3 - 0.5i - 1.3j + 1.4\hbar, & |p_1(t)|_1 > 2, \end{cases}$$

$$c_{22}(p_2(t)) = \begin{cases} -1.2 + 0.8i + 1.4j + 1.5\hbar, & |p_2(t)|_1 \le 2, \\ -1.2 + 0.8i + 1.6j + 1.3\hbar, & |p_2(t)|_1 > 2, \end{cases}$$

$$d_{11}(p_1(t)) = \begin{cases} 0.5 + 1.3i + 1.1j + 0.2\hbar, & |p_1(t)|_1 \le 2, \\ 0.4 + 1.3i + 1.0j + 0.2\hbar, & |p_1(t)|_1 > 2, \end{cases}$$

$$d_{12}(p_2(t)) = \begin{cases} -0.8 + 1.1i - 0.2j + 1.3\hbar, & |p_2(t)|_1 \le 2, \\ -1.1 + 1.1i - 0.2j + 1.1\hbar, & |p_2(t)|_1 > 2, \end{cases}$$

$$d_{21}(p_1(t)) = \begin{cases} -0.1 + 1.2i - 1.0j + 1.3\hbar, & |p_1(t)|_1 \le 2, \\ -0.1 + 1.1i - 1.2j + 1.1\hbar, & |p_1(t)|_1 > 2. \end{cases}$$

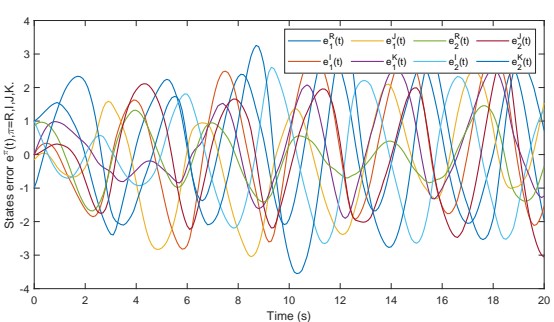

Fig. 1. The trajectories of error system states without controller.

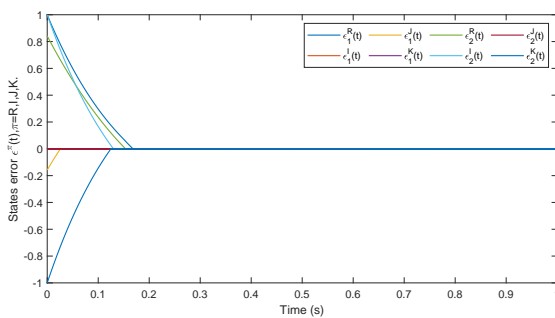

Fig. 2. The trajectories of error system states under the conditions in Theorem 1 by 2-norm.

Set $p_1(s) = -\cos(s) + (1 - \cos(s))i + \cos(s)j - \sin(s)\hbar$, $p_2(s) = -\sin(s) - \cos(s)i + 0.5\sin(s)j + (1 - \sin(s))\hbar$, $q_1(s) = \sin(s^2) + \sin(s^2)i + \sin(1-s)j - \sin(s^2)\hbar$, $q_2(s) = \sin(1-s) - \sin(s)i + 0.6\sin(s)j + \sin(s)\hbar$, where $s \in [-1, 0]$. It is demonstrated in Fig. 1 that the considered systems can not achieve FTS without controller.

Calculate $|\tilde{b}_{11}|_1 = 6$, $|\tilde{b}_{12}|_1 = 5.1$, $|\tilde{b}_{21}|_1 = 3.3$, $|\tilde{b}_{22}|_1 = 4.3$, $|\tilde{c}_{11}|_1 = 3.8$, $|\tilde{c}_{12}|_1 = 4.5$, $|\tilde{c}_{21}|_1 = 4.5$, $|\tilde{c}_{22}|_1 = 4.9$,

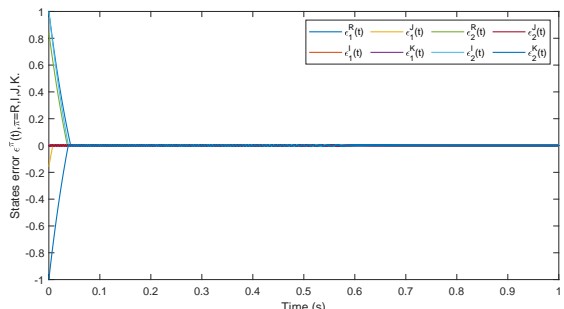

Fig. 3. The trajectories of error system states under the conditions in Theorem 2 by 1-norm.

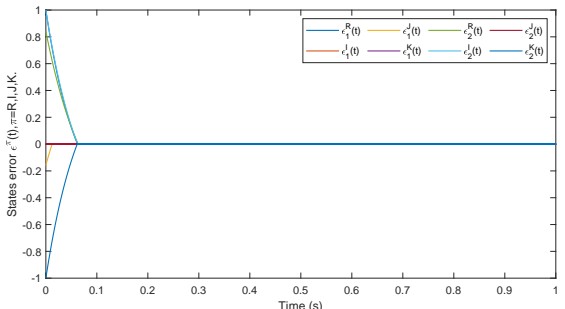

Fig. 4. The trajectories of error system states under the conditions in Theorem 3 by $\infty$-norm.

$|\tilde{d}_{11}|_1 = 3.1, |\tilde{d}_{12}|_1 = 3.5, |\tilde{d}_{21}|_1 = 3.6, |\tilde{d}_{22}|_1 = 3.7, |\tilde{b}_{11}|_2 = 3.03, |\tilde{b}_{12}|_2 = 2.75, |\tilde{b}_{21}|_2 = 2.51, |\tilde{b}_{22}|_2 = 2.27, |\tilde{c}_{11}|_2 = 2.41, |\tilde{c}_{12}|_2 = 2.36, |\tilde{c}_{21}|_2 = 2.36, |\tilde{c}_{22}|_2 = 2.52, |\tilde{d}_{11}|_2 = 1.79, |\tilde{d}_{12}|_2 = 1.92, |\tilde{d}_{21}|_2 = 2.03, |\tilde{d}_{22}|_2 = 2.04$, and select $\varepsilon = 0.5, \gamma_1 = 5.12, \gamma_2 = 4.77, \omega_1 = 4.9, \omega_2 = 4.2$. Therefore, conditions (5) and (6) in Theorem 1 are fulfilled, and the total synchronization time can be estimated as $6.10s$. In Fig. 2, the corresponding trajectories of error system (3) are shown, which indicate the achievement of the FTS under 2-norm.

Under the same initial conditions, let $\xi = 0.5, \gamma_1 = 12.3, \gamma_2 = 12.4, \omega_1 = \omega_2 = 18.9$. Then, it can be concluded that the conditions (8) and (9) are met, by Theorem 2, the FTS of QVMNNs (1) and (2) can be attained within $5.64s$, as depicted in Fig. 3.

Select $\xi = 0.5, \gamma_1 = 15.2, \gamma_2 = 13.4, \omega_1 = 10.9$, and $\omega_2 = 8.9$. Then, the synchronization time can be inferred as $3.59s$. Fig. 4 shows the applicability of Theorem 3.

**Remark 5.** In this example, the time-varying delay is selected as $\tau(t) = |\sin(t)|$, which fails to satisfy differentiability and $\dot{\tau}(t) < 1$. The simulation results indicate that the obtained theories are applicable to more general discrete time delays.

**Example 2.** To show the application of QVMNNs in secure communication, consider the following drive QVMNN with adaptive tracking signal

$$\dot{p}_r(t) = -a_r p_r(t) + \sum_{l=1}^{2} b_{rl}(p_l(t)) f_l(p_l(t))$$

$$+ \sum_{l=1}^{2} c_{rl}(p_l(t)) f_l(p_l(t - \tau(t)))$$

$$+ \sum_{l=1}^{2} d_{rl}(p_l(t)) \int_{t-\tau}^{t} f_l(p_l(s)) ds + \kappa_r(\chi_r(t) - \sigma_r(t)),$$

$$\dot{\sigma}_r(t) = \kappa_r(\chi_r(t) - \sigma_r(t)), \tag{12}$$

$r = 1, 2$, where $\kappa_r > 0, \chi_r(t), \sigma_r(t) \in \mathbb{Q}$ are the plaintext signal and the adaptive tracking signal, respectively. The other parameters are consistent with those in Example 1.

In this secure communication scheme, the sender encrypts the plaintext signal using the state information of the drive system (12) according to $\rho_r(t) = \chi_r(t) + p_r(t)$. The sender securely deletes the initial system values to prevent disclosure and transmits the system parameters, including $\kappa_r$ along with the ciphertext to the receiver. Upon receiving this information, the receiver constructs the response system as follows:

$$\dot{q}_r(t) = -a_r q_r(t) + \sum_{l=1}^{2} b_{rl}(q_l(t)) f_l(q_l(t))$$

$$+ \sum_{l=1}^{2} c_{rl}(q_l(t)) f_l(q_l(t - \tau(t)))$$

$$+ \sum_{l=1}^{2} d_{rl}(q_l(t)) \int_{t-\tau}^{t} f_l(q_l(s)) ds + u_r(t)$$

$$+ \kappa_r(\rho_r(t) - q_r(t) - \varrho_r(t)),$$

$$\dot{\varrho}_r(t) = \kappa_r(\rho_r(t) - q_r(t) - \varrho_r(t)), \tag{13}$$

$r = 1, 2$, where $\varrho_r(t)$ is the adaptive tracking signal. The receiver decrypts the ciphertext using $\hat{\rho}_r(t) = \rho_r(t) - q_r(t)$, Successful synchronization implies $\hat{\rho}_r(t) = \chi_r(t)$, ensuring the recovered plaintext matches the original. Figure 5 illustrates the secure communication scheme based on QVMNNs (12) and (13).

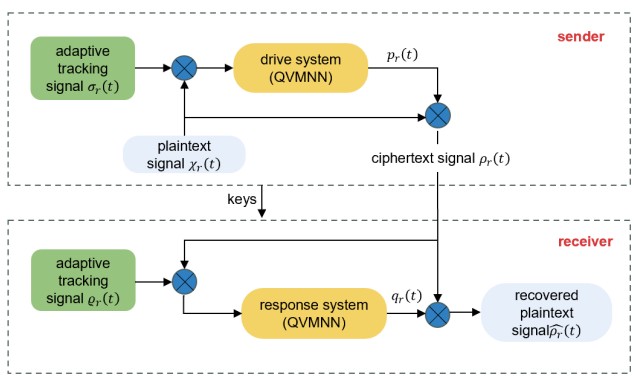

Fig. 5. The secure communication scheme.

Define $\check{\epsilon}_r(t) = q_r(t) - p_r(t)$, and $\hat{\epsilon}_r(t) = \varrho_r(t) - \sigma_r(t)$, then

$$\dot{\check{\epsilon}}_r(t) = R_r(t) - \kappa_r(\check{\epsilon}_r(t) + \hat{\epsilon}_r(t)),$$

$$\dot{\hat{\epsilon}}_r(t) = -\kappa_r(\check{\epsilon}_r(t) + \hat{\epsilon}_r(t)),$$

where $R_r$ equals to the right side of (3) for $n = 2$. Let $\hat{V}(t) = \frac{1}{2}\sum_{r=1}^{2}\check{\epsilon}_r^*(t)\check{\epsilon}_r(t) + \hat{\epsilon}_r^*(t)\hat{\epsilon}_r(t)$. Under the conditions of Theorem 1, it is straightforward to verify $D^+V(t) \leq 0$, indicating that the drive-response systems (12) and (13) can achieve synchronization, allowing the receiver to obtain the recovered plaintext.

Selecting $\kappa_1 = \kappa_2 = 1$, $\chi_1(t) = 0.2\sin(t) + 0.7\cos(t) + (3\sin(t) - 1.5\cos(t))i + (\sin(0.8t) + 0.3\cos(t))j + (\sin(t) + \cos(t))\hbar$, $\chi_2(t) = \sin(1.2t) - 0.5\cos(t) + 0.9\sin(t)i + (\sin(2t) - 0.2\cos(t))j + (\sin(1.5t) - \cos(t))\hbar$, $\sigma_1(s) = \sigma_2(s) = \varrho_1(s) = \varrho_2(s) = 0$, $s \in [-1, 0]$. and using the initial conditions and parameters from Example 1, Figures 6 and 7 illustrate the plaintext, ciphertext, and recovered plaintext signals.

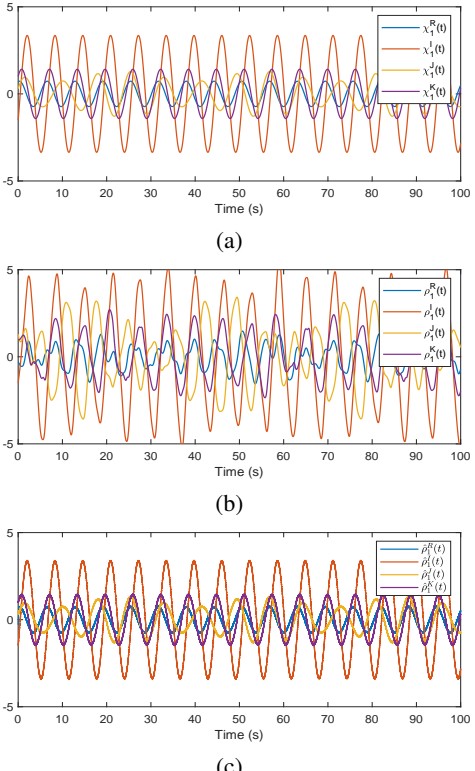

Fig. 6. Secure communication (a) Plaintext $\chi_1(t)$. (b) Ciphertext $\rho_1(t)$. (c) Recovered plaintext $\hat{\rho}_1(t)$.

## V. CONCLUSIONS

This paper explores the FTS control of QVMNNs with discrete and distributed time delays by Halanay inequality. For the error system, distinct differential inequalities are formulated to accommodate states inside and outside the unit ball. By employing the Halanay inequality and finite-time stability theory, algebraic criteria are established to achieve FTS of the QVMNNs under the influence of a controller composed solely of linear and sign function terms. Notably, the introduction of 1-norm, 2-norm, and $\infty$-norm of quaternion facilitates the construction of Lyapunov functions using simple norm forms,

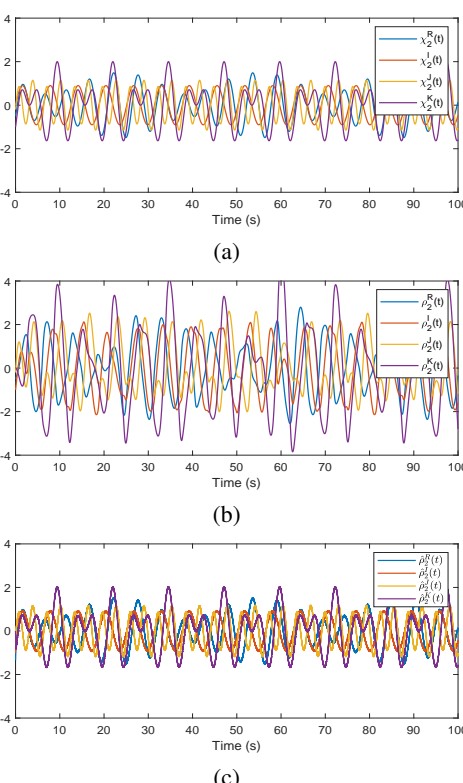

Fig. 7. Secure communication (a) Plaintext $\chi_2(t)$. (b) Ciphertext $\rho_2(t)$. (c) Recovered plaintext $\hat{\rho}_2(t)$.

thereby enabling the analysis of QVMNNs through a non-decomposition method. Besides, estimates of the synchronization time are provided for different initial conditions. Given that the robustness of systems is critical when coping with uncertainty and unforeseen circumstances, our future research will endeavor to study the robustness of QVMNNs with mixed time delays.

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
