# OpenReview forum: "Finite-Time Synchronization Control for Quaternion-Valued Memristive Neural Networks by Halanay Inequality"
_IEEE.org/ICIST/2024/Conference — IEEE ICIST 2024 Conference Submission_

### Official Review · Reviewer_yvBV · 2024-08-29
**This paper can be accepted.**

**Rating:** 9
**Confidence:** 4

**Review:**

This paper explores the FTS control of QVMNNs with discrete and distributed time delays by Halanay inequality. The FTS criteria for QVMNNs under 1-norm, 2-norm, and ∞-norm are given. The effectiveness of the method is shown through simulation results.

This paper is well written and the results are positive. The derivations seem correct. It is suggested that the comparisons with existing results should be given through simulations.

---

### Official Review · Reviewer_ZgpK · 2024-08-29
**comment**

**Rating:** 7
**Confidence:** 5

**Review:**

This paper investigates the finite-time synchronization (FTS) of quaternion-valued memristive neural networks (QVMNNs) with mixed time delays. The reviewer has the following comments:
1.	The differences of using 1-norm, 2-norm, and infinity-norm of quaternion should be clarified.
2.	Why Dini derivative of Lyapunov function is used in this work?
3.	Some langrage issue should be corrected.

---

### Official Review · Reviewer_6pMd · 2024-08-30
**Review on Finite-Time Synchronization Control for Quaternion-Valued Memristive Neural Networks by Halanay Inequality**

**Rating:** 8
**Confidence:** 3

**Review:**

1. What is Halanay inequality?
2. What is the meaning of the synchronization of QVNNs? Some specific applications are needed to be supplemented.
3. The results of simulation example 2 are not well illustrated.

---

### Decision · Program_Chairs · 2024-09-06

Accept (Oral)